# Genome-Wide Identification and Expression Analysis of the *Aux/IAA* Gene Family of the Drumstick Tree (*Moringa oleifera* Lam.) Reveals Regulatory Effects on Shoot Regeneration

**DOI:** 10.3390/ijms232415729

**Published:** 2022-12-11

**Authors:** Endian Yang, Heyue Yang, Chunmei Li, Mingyang Zheng, Huiyun Song, Xuan Zou, Xiaoyang Chen, Junjie Zhang

**Affiliations:** 1College of Forestry and Landscape Architecture, South China Agricultural University, Guangzhou 510642, China; 2Guangdong Key Laboratory for Innovative Development and Utilization of Forest Plant Germplasm, Guangzhou 510642, China; 3State Key Laboratory for Conservation and Utilization of Subtropical Agro-Bioresources South China Agricultural University, Guangzhou 510642, China; 4Guangdong Province Research Center of Woody Forage Engineering Technology, Guangzhou 510642, China

**Keywords:** *Moringa oleifera* Lam., auxin, Aux/IAA gene family, shoot regeneration

## Abstract

Auxin plays a critical role in organogenesis in plants. The classical auxin signaling pathway holds that auxin initiates downstream signal transduction by degrading Aux/IAA transcription repressors that interact with ARF transcription factors. In this study, 23 *MoIAA* genes were identified in the drumstick tree genome. All *MoIAA* genes were located within five subfamilies based on phylogenetic evolution analysis; the gene characteristics and promoter cis-elements were also analyzed. The protein interaction network between the MoIAAs with MoARFs was complex. The *MoIAA* gene family responded positively to NAA treatment, exhibiting different patterns and degrees, notably for *MoIAA1*, *MoIAA7* and *MoIAA13*. The three genes expressed and functioned in the nucleus; only the intact encoding protein of *MoIAA13* exhibited transcriptional activation activity. The shoot regeneration capacity in the *35S::MoIAA13-OE* transgenic line was considerably lower than in the wild type. These results establish a foundation for further research on *MoIAA* gene function and provide useful information for improved tissue culture efficiency and molecular breeding of *M. oleifera*.

## 1. Introduction

*Moringa oleifera* Lam. is a fast-growing tree from southern Asia and has since spread throughout the world’s tropical and subtropical regions; it is commonly known as the drumstick tree [1]. *M. oleifera* has gained attention because of its rich nutritional composition [2], potential medicinal applications [3], and considerable industrial value [4]. In previous studies, we successfully established an induction system for drumstick trees using vitro young leaves of sterilized seedlings [5]. However, the induction rate is genotype-dependent. Furthermore, we found that hormones had a significant effect on drumstick tree shoot regeneration, especially auxin. A study of auxin early response genes has revealed that the *Aux/IAA (Auxin/Indole-3-Acetic)*, *ARF (Auxin Response Factor)*, *GH3 (Gretchen Hagen 3),* and *SAUR (Small Auxin Up-regulated RNA)* family genes are closely involved in the embryogenic transformation of upland cotton callus; Aux/IAAs, in particular, display a high level of expression [6]. Global transcriptomic analysis implied that *Aux/IAA* gene family is involved in the molecular regulatory network during drumstick tree shoot regeneration in our previous research [7].

The *Aux/IAA* family comprises auxin early response genes, responding to auxin levels rapidly and precisely to participate in cellular processes, such as cell differentiation and division [8]. As a repressor of auxin-inducible gene expression, the *Aux/IAA* family is pivotal in the auxin signaling pathway [9]. The increasing popularity of whole genome sequencing has led to *Aux/IAA* genes of several species being identified, including *A. thaliana* [10], *O. sativa* [11], *P. trichocarpa* [12] and *E. grandis* [13]. In the auxin signaling pathway, Aux/IAA proteins are a type of short-lived protein, repressing the activity of ARFs (auxin response factors) in the absence of auxin [14]. In contrast, at high levels of auxin, Aux/IAA proteins bind with SCF^TIR1/AFB^ and subsequent degradation via the ubiquitination pathway, thus activating ARF proteins and inducing downstream gene expression [15]. Aux/IAA proteins engage in the spatial-temporal distribution of auxin by regulating the auxin signaling pathway [16].

Aux/IAA proteins usually have four highly conserved domains, designated I-IV, that deliver distinct functions [10]. Domain I consists of a LxLxLx motif that interacts with TPL (TOPLESS) and is responsible for transcriptional repression [17]. Domain II is a key region for protein degradation with a GWPPV motif that interacts with SCF^TIR1/AFB^ [18]. Domains III and IV contain a Phox and Bem 1 (PB1 domain), which share homology with domains of ARFs, and thus are responsible for transcriptional repression between ARFs and other Aux/IAA proteins [19]. In recent years, molecular and genetic studies have uncovered the functions of Aux/IAA proteins [8]. These studies emphasized that *Aux/IAA* genes display diverse functions in developmental processes, including embryo development, root and shoot tropisms, and leaf patterns [20].

It has been confirmed that Aux/IAA proteins were involved in shoot regeneration by interacting with ARF proteins. IAA12 controls shoot regeneration in combination with ARF4 and ARF5 in *Arabidopsis* [21]. In this study, 23 *Aux/IAA* genes were identified and characterized in *M. oleifera*. The expression patterns of *MoIAA* genes were examined at three stages in shoot regeneration. Moreover, *MoIAA13* functional roles were investigated for the first time in *N. benthamiana*. Our findings provide useful information to help to unravel the mechanisms of the function of *Aux/IAA* in shoot regeneration, and establish the foundation for improved tissue culture efficiency and molecular breeding for *M. oleifera*.

## 2. Results and Discussion

### 2.1. Genome-Wide Identification of the Aux/IAA Gene Family of M. oleifera

The *Aux*/*IAA* genes of *Arabidopsis thaliana* were used to search for *Aux*/*IAA* genes in the genomes of *M. oleifera* using TBtools [22]. Their deduced peptides were confirmed after domain analysis using Pfam and NCBI-CDD online search platforms. A total of 23 members were finally found and designated *MoIAA1* to *MoIAA23* (Table 1). Compared with *A. thaliana* (29 *AtIAAs*), *O. sativa* (31 *OsIAAs*) and *P. trichocarpa* (35 *PopIAAs*), the number of *M. oleifera* IAAs is low (23 *MoIAAs*), but the same as *Prunus persica* (23 *PpIAAs*) [23]. The amino acid sequences of MoIAAs were further analyzed. The amino acid length ranged from 135 aa to 506 aa in MoIAAs (Table 1). The MWs ranged from 14.69–54.62 kDa, and the pI varied from 4.67 to 9.54 (Table 1). It was predicted that all MoIAAs were located in the nucleus. The characterization of MoIAAs has similarities with other plants [24].

### 2.2. Comprehensive Analysis of MoIAA Gene Structure

In a previous study, the phylogenetic analysis of 434 *Aux/IAA* genes in 17 plant species showed that the Aux/IAA proteins can be broadly categorized into five primary clades [18]. The 23 MoIAA’s protein sequences were also divided into five subfamilies (I, II, Ⅲ, IV and Ⅴ) by phylogenetic analysis (Figure 1A). We then used the alignment of genomic DNA with full-length cDNA of *MoIAAs* to explore structures of exons/introns. All of the *MoIAA* genes except *MoIAA1* had introns (lamu_GLEAN_10000198). Most members had a typical gene structure pattern of five exons and four introns; only *MoIAA17* (lamu_GLEAN_10014915) had one intron (Figure 1C). Meanwhile, the conserved motifs of MoIAAs were closely identified, and the MEME website found eight. The C-terminus was composed of motif 4, motif 6, motif 3, and motif 2, and the N-terminus consisted of motif 7, motif 1, motif 5 and motif 8. The C-terminus is conserved, and the N termini of Aux/IAA proteins interact with ubiquitin ligase receptors through motif-based interactions [25]. MoIAA proteins have four highly conserved domains, including domain I (motif 4), domain II (motif 3), domain Ⅲ (motif 2) and domain IV (motif 1) (Figure 1D). Recent studies have shown that deficiency in domains may lead to changes in protein properties, and the functions of individual domains in Aux/IAA proteins have been revealed [18]. Only 15 MoIAAs had four complete domains. Several motifs were common to all MoIAAs, such as motif 2 (domain Ⅲ). Domain III was the PB1 domain, which is a dimerization region involved with other PB1 domain proteins, such as ARFs [26]. In addition, members of the same subfamily showed generally similar gene structures, which suggested they were highly conserved (Figure 1B).

Furthermore, we undertook a multiple alignment analysis of the 23 MoIAA protein sequences in the drumstick tree. Most of the MoIAAs were found to contain nuclear localization signals (NLSs), indicating that the nuclear localization signal in Aux/IAA proteins is conserved, and Aux/IAAs are probably localized in the nucleus [24]. The motif (one β sheet and two α helices) acting in Aux/IAA dimerization was also found in domain III of most MoIAAs. Meanwhile, some members were missing domains; for example, MoIAA1 and MoIAA3 were without domain IV (VKVxMxG), and MoIAA17 and MoIAA18 were without domain I (LxLxL). A similar pattern of results was obtained in different plant species; for example, the Aux/IAA proteins in tomato (SlIAA13 SlIAA16 and SlIAA20) and in peach (PpIAA26 and PpIAA33) also lack domain I [27]. Meanwhile, there is a mutation of domain I in AtIAA7, AtIAA17 and AtIAA19, which partially reduces the inhibition of auxin-modulated transcription [28]. “V” was mutated to “I” in domain II (GWPPV) in some MoIAAs (Figure 2). In addition, MoIAA9, MoIAA16 and MoIAA17 were without domain II, implying that these proteins may not be quickly degraded at high auxin levels.

### 2.3. Phylogenetic and Evolution Analysis of MoIAAs

Previous studies have revealed that phylogenetic analysis can help elucidate evolutionary relationships as well as predict the potential functions of various genes [29]. Here, all Aux/IAA proteins in *M. oleifera* (23 MoIAAs), *A. thaliana* (29 AtIAAs), *O. sativa* (31 OsIAAs), *P. trichocarpa* (35 PoptrIAAs) and *E. grandis* (24 EgrIAAs) were compared using ClustalW. A phylogenetic tree was generated with the maximum likelihood method to reveal their evolutionary relationships (Figure 3). All the Aux/IAA protein sequences are listed in Appendix A. All the Aux/IAA proteins were classified into five subfamilies, including 10 groups (Groups A to J). This was similar to the phylogenetic analysis of the *Eucalyptus Aux/IAA* gene family, which is divided into 11 groups [13]. There were slightly fewer Aux/IAA proteins in *M. oleifera*—only 23. MoIAAs were present in all groups except B (Figure 3). The highest number of members were placed in group J, and these have four complete domains. Group H had only one *M. oleifera* member, MoIAA1 (lamu_GLEAN_10000198), which has a higher homology than EgrIAA19. MoIAA14 (lamu_GLEAN_10011220) has a close relationship with EgrIAA4, and a previous study revealed that overexpressing EgrlIAA4 in *Arabidopsis* impaired its growth and auxin-sensitive ability, suggesting that MoIAA14 may also impede the auxin response ability. MoIAA15 (lamu_GLEAN_10013390) has higher homology to EgrIAA9A and PoptrIAA9, implying MoIAA15 is also involved in wood formation [15].

Moreover, MoIAAs are in a different branch than OsIAAs, indicating that *M. oleifera* is not closely related to *O. sativa* (Figure 3). During plant evolution, gene expansion and duplication events often occur among members of the same gene family, which make gene functions diverse and specific [30]. It has been demonstrated that segmental and tandem duplications occur most frequently in plants, and these have played a crucial role in expanding gene families [31]. We detected only two pairs of tandemly duplicated genes in the 23 genes of the *MoIAA* gene family, namely *MoIAA13*-*MoIAA14* and *MoIAA21*-*MoIAA22*. Most gene pairs were identified as segmentally duplicated genes; it appears that segmental duplication events were the main cause of the expansion of the *Aux/IAA* gene family in *M. oleifera*. The Ka/Ks ratios of duplicate gene pairs (Table 2) were all less than 0.5, suggesting that *MoIAAs* were under strong purifying selection. These basic results are consistent with a previous study reporting that *MdAux/IAAs* had undergone purifying selection [32], implying the *Aux/IAA* genes tended to lose mutations by purifying selection because they adapted to their current environment.

### 2.4. Promoter Cis-Element Analysis

Several studies have indicated that promoters influence the temporal and spatial expression of genes, and cis-elements within promoters are key to regulating gene function through interplay with trans-acting factors [33]. Multiple *Aux/IAA* genes with promoter elements associated with abiotic stress and hormones have been found to participate in these abiotic stress responses [34]. For example, *OsIAA20* and most of the *MtIAA* genes exhibited wide-ranging responses to salt stress, and more than half of the *FveIAA* genes responded to IAA treatment [35]. The promoter analysis of *MoIAAs* detected nine major cis-elements, including low-temperature-responsive, light-responsive and hormone-responsive elements (Figure 4). Auxin-responsive and light-responsive related elements were abundant. In addition, there were also more MYB binding sites, indicating that *Aux/IAA* genes may be related to MYB transcription factors and light-induced plant growth regulation. There is evidence that *CpAux/IAA* gene expression decreases when in vitro *Carica papaya* plantlets are exposed to high light intensity, even though no auxin has been added [36]. Notably, a wound-responsive element was identified in the *MoIAA17* (lamu_GLEAN_10014915) gene promoter, which may be involved in the dedifferentiation process induced by wound signaling. Furthermore, meristem-expression and cell-cycle-regulation elements were also identified, suggesting that *MoIAAs* may be involved in regulating cell division.

### 2.5. Predicted MoIAA and MoARF Protein Interaction Network

It has been found that the Aux/IAA proteins negatively regulate the ARF proteins, so there is a high potential for interaction between them. A total of 19 ARF members were identified in *M. oleifera* (Appendix A). Therefore, the protein–protein interaction network for the 23 MoIAA and 19 MoARF proteins was constructed using the STRING online tool [37]. The results indicated that 17 MoIAAs and 13 MoARFs interact closely (Figure 5). Generally, the interaction between MoIAAs and MoARFs was complex; some MoIAA proteins may interact with several MoARFs and vice versa, representing functional redundancy within the Aux/IAA proteins [38]. For example, MtIAA12 and MtIAA21 interacted with MtARF29, which also interacts with most of the MtARFs [34]. Notably, MoIAA3 and MoIAA13 proteins were at the center of the network, suggesting that they interact strongly with other proteins, while MoARF2.1 and MoARF17 exhibit the least interaction with others. Therefore, MoIAA3 and MoIAA13 may be the key proteins in regulating the auxin signaling pathway by interacting with other MoIAA and MoARF proteins.

### 2.6. Expression Analysis of MoIAA Genes in Response to NAA Treatment

Auxin is essential for callus formation in isolated tissues of plants. As a primary auxin-responsive gene family, *Aux/IAA* genes respond quickly to auxin treatment to regulate downstream genes [39]. Previous studies have focused on tissue expression patterns and plant development of Aux/IAA genes [40]. In this study, to understand how the *MoIAAs* respond to NAA treatment in callus induction and shoot regeneration of *M. oleifera*, the leaf explants were inoculated with different NAA concentrations. The leaves were then collected after 0 h, 6 h, 12 h, and 24 h to study all *MoIAA* gene expression patterns using qRT-PCR. When compared to the expression of these genes in untreated controls, we discovered that NAA stimulated most MoIAA genes. However, NAA had little effect on the gene expression of *MoIAA4* (lamu_GLEAN_10004873) and *MoIAA21* (lamu_GLEAN_10018587), which remained steady (Figure 6). With the rise in NAA concentration, the expression of some members increased across all time points, while the expression levels of almost all members remained constant over time with 0 mg/L NAA (Appendix A). Of the entire *MoIAA* gene family, *MoIAA1* (lamu_GLEAN_10000198) and *MoIAA7* (lamu_GLEAN_10007551) have the highest expression, responding strongly to NAA (Figure 6). *MoIAA1* expression was dramatically upregulated after NAA treatment, increasing nearly 50-fold in just 6 h.

Moreover, *MoIAA7* expression reached its highest value after 6 h of NAA treatment, then continued to decline, and the expression level had a corresponding relationship with NAA concentration. In contrast, the expression of *MoIAA13* (lamu_GLEAN_10011219) increased with increasing treatment time (Figure 7). *IAA11* and *IAA14* gene expression exhibits a general downward trend throughout the embryogenic callus induction process in Tamarillo, while IAA17 expression appears greater than that in the initial explant [41]. Other research has revealed that shoot regeneration ability was hampered by increased *IAA12* expression [21]. These results showed that the *MoIAA* gene family responds positively to NAA treatment, exhibiting different patterns and to different degrees, indicating that *MoIAAs* are involved in the regulation of adventitious shoot regeneration.

### 2.7. Subcellular Location of MoIAA1, MoIAA7 and MoIAA13 Proteins

Aux/IAA proteins act as transcription factors in the nucleus [42]. Full-length coding sequences from *MoIAA1*, *MoIAA7* and *MoIAA13* genes were cloned to validate the protein subcellular location prediction results. The three *MoIAA* genes without a stop codon were then cloned into the pEarleyGate101 vector with the YFP protein (Appendix A). The results of subcellular localization revealed that YFP fused separately with these three MoIAA proteins are expressed in the nucleus (Figure 8), proving that the *MoIAA1*, *MoIAA7* and *MoIAA13* genes are functional in the nucleus.

### 2.8. Validation of MoIAA1, MoIAA7 and MoIAA13 Transcriptional Activation Activity

To verify the transcriptional activation activity of the *MoIAA1*, *MoIAA7* and *MoIAA13* encoding proteins, the CDSs of the three genes were ligated into the pGBKT7 vector. The vectors were transformed into yeast cells using the PEG/LiAc-mediated transformation method [43]. The results showed that the yeasts transformed with the empty vector, and the recombinant vector grew normally on SD/-Trp solid medium. However, the yeasts transformed with pGBKT7 (empty vector), pGBKT7-*MoIAA1*, and pGBKT7-*MoIAA7* do not grow properly on the medium with SD/-Trp-His and X-α-gal, while the yeast transformed with the pGBKT7-*MoIAA13* grows normally and turns blue (Figure 9). It was demonstrated that only the intact *MoIAA13* encoding protein exhibits transcriptional activation activity in these three genes.

### 2.9. MoIAA13 Regulates the Process of Shoot Regeneration

The special expression patterns of *MoIAA1*, *MoIAA7* and *MoIAA13* in association with NAA treatment suggest their involvement in adventitious shoot regeneration. Further experiments revealed that they were all expressed in the nucleus, but only *MoIAA13* shows transcriptional activation activity. Consequently, we are interested in the effect of *MoIAA13* on adventitious shoot regeneration. We established transgenic lines of Overexpression *MoIAA13* in *N. benthamiana* to investigate the shoot regeneration capacity. We found that the regeneration frequency in the *35S::MoIAA13-OE* transgenic line was considerably lower than in the wild type (Figure 10). The occurrence of adventitious buds is dependent on the formation of primordia resembling the lateral root primordia (LRP) [44]. The specification of founder cells is the most important event in pluripotent primordia formation. Cell specification within cells with local auxin maxima will act as founder cells and subsequently develop into pluripotent primordia [45]. Therefore, the expression of genes related to the auxin signaling pathway is crucial for founder cell specification and primordia formation [46]. There is a population of stem cells in the primordia that can be induced to differentiate by inducing changes in gene expression [47]. Several phytohormones deliver a specific molecular signal and play key roles in plant cell division and differentiation [7]. Auxin works by directly binding to the nuclear receptor of TIR1/AFB proteins; the auxin-TIR1/AFB co-receptors inhibit the *Aux/IAA* transcriptional repressors. The release of *ARF* transcription factors leads to the activation of downstream genes [46]. *Aux/IAAs* and *ARFs* are important regulators of all stages of shoot regeneration. *ARFs* activate the initiation of the LRP by activating the transcription factor of *LBDs* (Lateral Organ Boundaries Domains), while *ARFs* are negatively regulated by *Aux/IAAs*. The *MoLBD* gene family was found to be differently expressed at the callus differentiation stage in our previous study [7]. *ARF7* activates *SAUR19* expression by binding its promoter directly, and several *SAUR* genes have been found to exhibit highly increased expression across the coffee somatic embryogenesis process [48]. There are studies revealing that Auxin-induced callus formation is retarded when the repressive interaction of *IAA19* with *ARF7* is destabilized [49]. *IAA12* overexpression hampered shoot regeneration capacity, which can be complemented by *ARF4* overexpression, and *ARF4* regulates shoot regeneration by collaborating with *ARF5* and *IAA12* [21]. In this study, our results are consistent with these findings, implying that *MoIAA13* interacts with *MoARFs* in response to auxin signaling, thereby regulating shoot regeneration. It is expected that future studies will reveal the specific details of this signaling mechanism.

## 3. Material and Methods

### 3.1. Identification of the Aux/IAA Genes in Drumstick Tree

The complete protein sequences of *Moringa oleifera* Lam. are available from NCBI under the accession number PRJNA268707. We used *Arabidopsis thaliana Aux/IAA* proteins [10] as queries in BLASTP searches with an E value lower than 1 × 10^−10^ for predicted proteins in the drumstick tree genome. The candidate proteins were then verified for the presence of the Aux/IAA domain using NCBI-CDD [50]. Ultimately, 23 *Aux/IAA* protein sequences of the drumstick tree were obtained.

### 3.2. Analysis of Physicochemical Properties of Amino Acids

The number of amino acids, molecular weight, and theoretical isoelectric points of all *MoIAAs* were analyzed using the online ProtParam tool provided by ExPASY (https://web.expasy.org/protparam/) (accessed on 31 January 2022). Then, their subcellular locations were predicted using Plant-mPLo (http://www.csbio.sjtu.edu.cn/bioinf/plant-multi/) (accessed on 31 January 2022).

### 3.3. Gene Structure and Phylogenetic Analysis

The structures of *MoIAAs* were visualized using Tbtools software, with the gene annotation described in the GFF3 format [22]. Motifs of the protein sequences were investigated with the MEME tool (http://meme-suite.org/index.html) (accessed on 31 January 2022). The multiple alignments of protein sequences were performed using the ClustalW function of MEGA 7., and visualized with the Jalview software [51]. Phylogenetic relationships were established using MEGA 7.0 by the maximum likelihood (ML) method [51]. A non-parametric bootstrapping was performed with 1000 replicates.

### 3.4. Analysis of Aux/IAA Gene Duplication and Evolution

MCScanX software was used to analyze the gene duplication and syntenic events based on full-length MoIAA sequences with the default parameters [52]. The Ka/Ks ratio between gene pairs was calculated using TBtools software [22]. A Ka/Ks ratio less than 1 indicates ‘‘negative selection’’ and greater than 1 represents ‘‘positive selection’’ [53].

### 3.5. Analysis of Cis-Acting Elements in Aux/IAA Gene Promoters

The 2.0-kb upstream sequences from the translation start sites of the *MoIAAs* were extracted from the drumstick tree genome, then submitted to PlantCARE (http://bioinformatics.psb.ugent.be/webtools/plantcare/html/) (accessed on 31 January 2022). The cis-acting elements were visualized using TBtools software [22].

### 3.6. Predicted Protein–Protein Interaction Network

First, we used AtARF proteins as queries for the predicted MoARF proteins in the drumstick tree genome. Then, analysis of the interaction network of the 23 MoIAAs and 19 MoARFs was performed using the STRING functional protein association network (https://string-db.org) (accessed on 8 September 2022), calculated and visualized by Cytoscape software [54].

### 3.7. Expression Analysis

The leaf explants with consistent growth status were inoculated onto MS medium containing NAA concentrations of 0 mg/L, 0.1 mg/L, 0.5 mg/L, and 1 mg/L. Samples were taken at three time points—6 h, 12 h, and 24 h—and three biological replicates were examined. The leaves treated at 0 h were used as the control (CK). These samples were frozen in liquid nitrogen and then stored at −80 °C. The forward primers and reverse primers listed in Appendix A were designed using Primer 5.0. ACP2 was amplified as a reference gene [55]. Gene expression was detected with RT-qPCR, and the relative expression level for each gene was estimated using 2^−ΔΔCT^ [7].

### 3.8. Subcellular Localization

The coding sequences without the stop codon of *MoIAA1* (lamu_GLEAN_10000198), *MoIAA7* (lamu_GLEAN_10007551) and *MoIAA13* (lamu_GLEAN_10011219) were cloned into the pEarley Gate101 vector in a frame with EYFP and the CaMV 35 S promoter. The recombinant vector was transformed into *Agrobacterium* (GV3101) for the transformation of *N. benthamiana*. Then, subcellular localization was analyzed by *N. benthamiana* leaves [56]. The YFP signals were detected using a fluorescence microscope.

### 3.9. Transactivation Activity Analysis in Yeast

The full-length coding sequences of *MoIAA1*, *MoIAA7* and *MoIAA13* were amplified using the primers listed in Appendix A and cloned into the pGBKT7 vector. The recombinant and control vectors were transformed into the yeast strain Y2HGold [43]. Then, the transformed strains were cultured on SD/-Trp and SD/-Trp/-His/x-α-Gal media at 30 °C for four days.

### 3.10. Generation of Overexpressing MoIAA13 Transgenic N. benthamiana Lines

The full-length coding sequences of *MoIAA13* were cloned into the pEarley Gate101 vector, which is a binary vector with a CaMV 35S promoter. Then, the empty vector and recombinant vector were transferred into *Agrobacterium* (GV3101) for the transformation of *N. benthamiana.* The transgenic plants were identified by PCR amplification and qPCR. The leaf explants of WT, the empty vector, and the *35S::MoIAA13*-*OE* transgenic lines were inoculated on MS medium containing 1.0 mg/L 6-BA and 0.1 mg/L NAA. After 25 days, the regeneration frequency (ratio of explants with regenerated buds to total explants) was determined from three independent replicates containing 30 explants. The data presented are mean ± SD from the replicates.

## 4. Conclusions

This study identified and characterized the *Aux/IAA* gene family of *M. oleifera*. Bioinformatics analysis suggested the *MoIAAs* were highly conserved. Meanwhile, the structure of MoIAA proteins was different, indicating the changes in protein properties and functions. Auxin-responsive and abiotic stress-related elements were detected in the *MoIAA* genes promoter. MoIAA and MoARF protein interaction network predicted analysis indicated that they interact closely. The results of the preliminary experimental analysis demonstrated that the *MoIAA13s* were sensitive to NAA treatment and exhibited transcriptional activation activity, and may have significant effects on shoot regeneration capacity during plant tissue culture. The findings provide a reference for future functional studies of this gene and establish the foundation for improved tissue culture efficiency and molecular breeding for the drumstick tree.

## Figures and Tables

**Figure 1 ijms-23-15729-f001:**
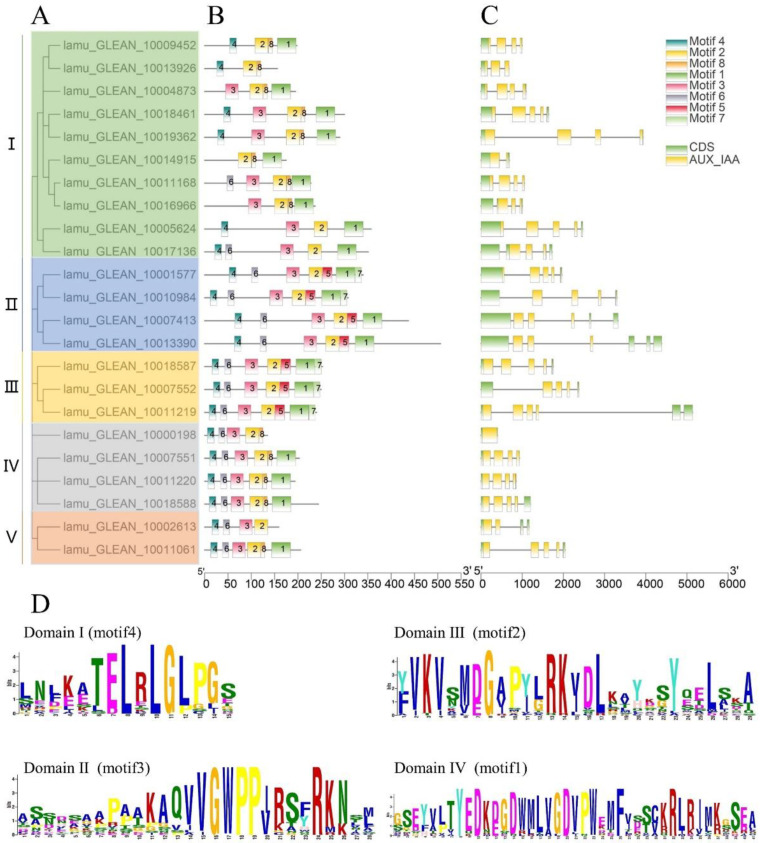
Gene structure and motif analysis of MoIAA proteins from *M. oleifera* (**A**). The phylogenetic tree of all MoIAA proteins (**B**). The motifs in the MoIAA proteins (**C**). The exon–intron structure distribution of MoIAAs (**D**). The abscissa of sequence logos refers to the amino acid with the highest frequency, and the ordinate represents the relative frequency of the corresponding amino acid.

**Figure 2 ijms-23-15729-f002:**
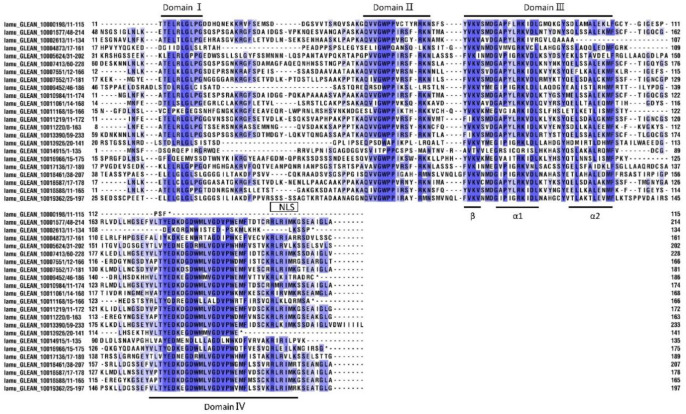
Multiple sequence alignment of conserved domains in MoIAAs’ proteins.

**Figure 3 ijms-23-15729-f003:**
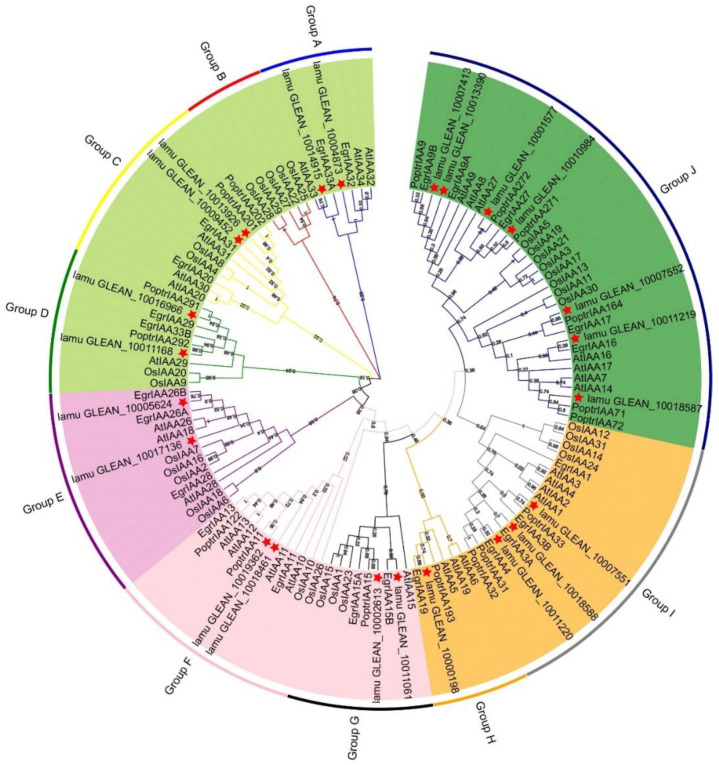
Phylogenetic tree of the Aux/IAA proteins family for *M. oleifera*, *A. thaliana*, *O. sativa*, *P. trichocarpa*, and *E. grandis*. The phylogenetic tree was constructed with 1000 bootstrap replications in MEGA7.0. Five colors represent the five subfamilies, and a red star indicates MoIAAs.

**Figure 4 ijms-23-15729-f004:**
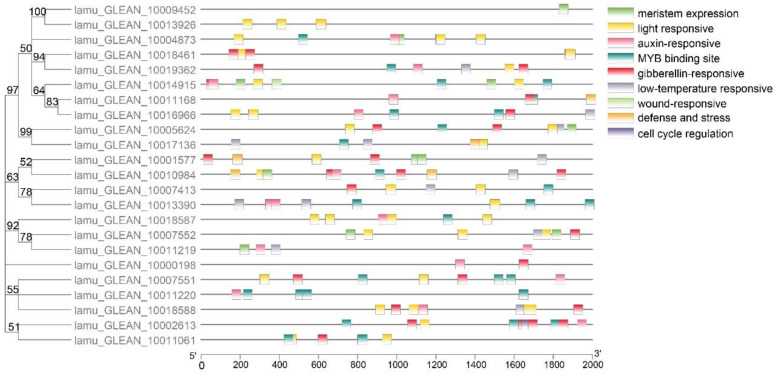
Prediction and visualization of cis-elements in *MoIAA* promoters.

**Figure 5 ijms-23-15729-f005:**
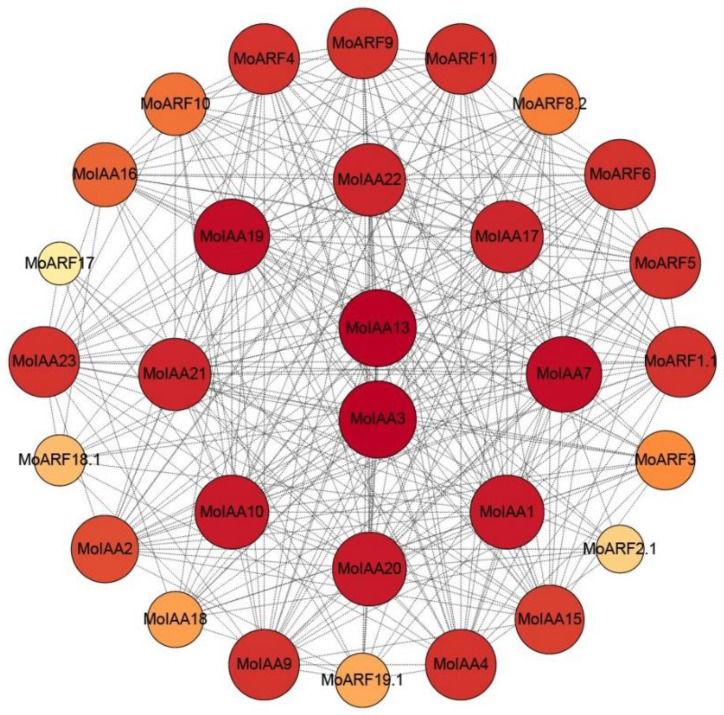
Protein–protein interaction network for the MoIAA and MoARF gene family; the network contains 30 nodes, and darker colors and larger nodes represent more interaction with others.

**Figure 6 ijms-23-15729-f006:**
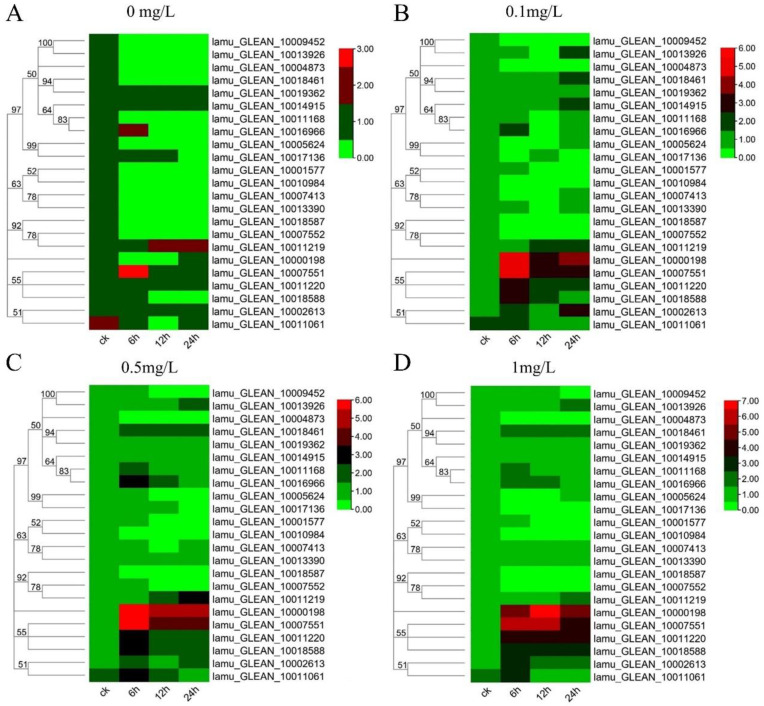
Heatmap of *MoIAA* gene expression level under 0 mg/L (**A**), 0.1 mg/L (**B**), 0.5mg/L (**C**), 1.0 mg/L (**D**) NAA treatment. The scale represents the logarithm of the value of gene expression.

**Figure 7 ijms-23-15729-f007:**
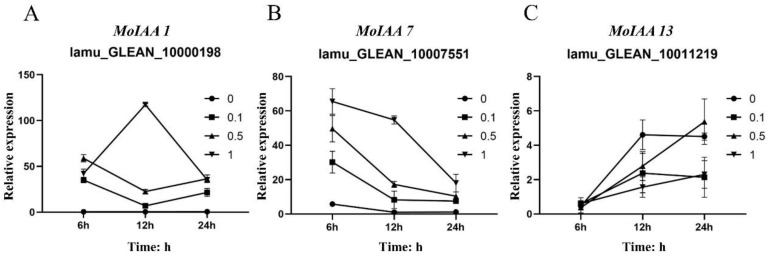
Relative expression level of *MoIAA1* (**A**), *MoIAA7 (***B***)* and *MoIAA13* (**C**) under NAA treatment at 6, 12, and 24 h. The expression value was calculated relative to 0 h.

**Figure 8 ijms-23-15729-f008:**
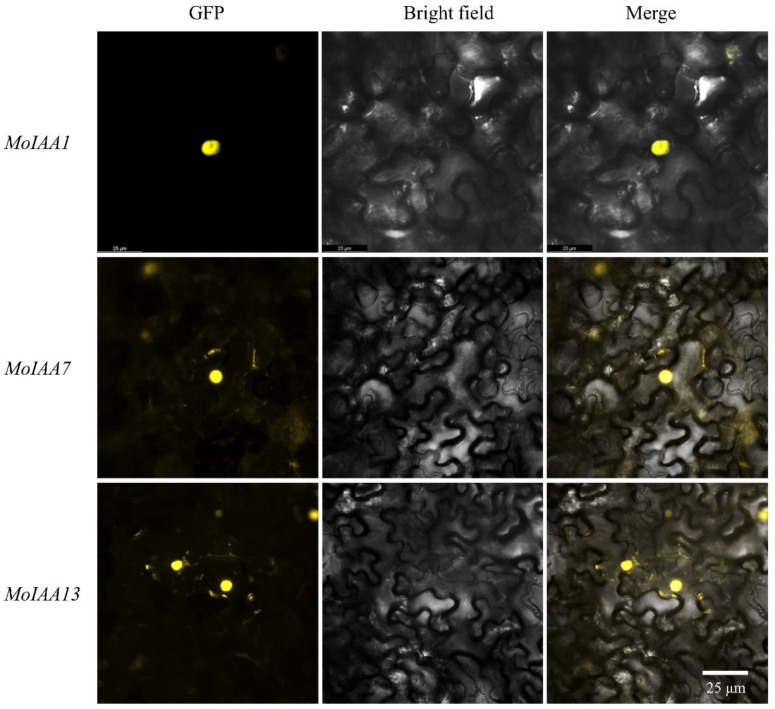
Subcellular localization of the *MoIAA1*, *MoIAA7* and *MoIAA13* proteins in *N. benthamiana* leaves.

**Figure 9 ijms-23-15729-f009:**
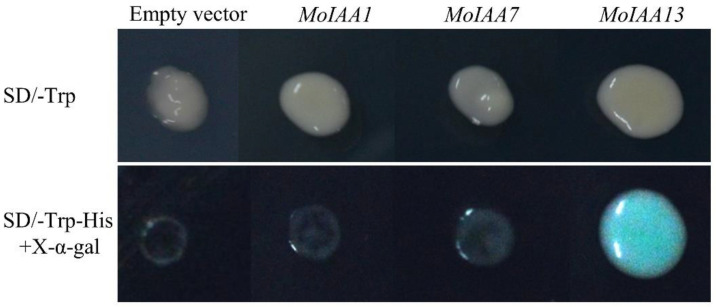
Analyses of transcriptional activation activity of the *MoIAA1*, *MoIAA7* and *MoIAA13* encoding proteins.

**Figure 10 ijms-23-15729-f010:**
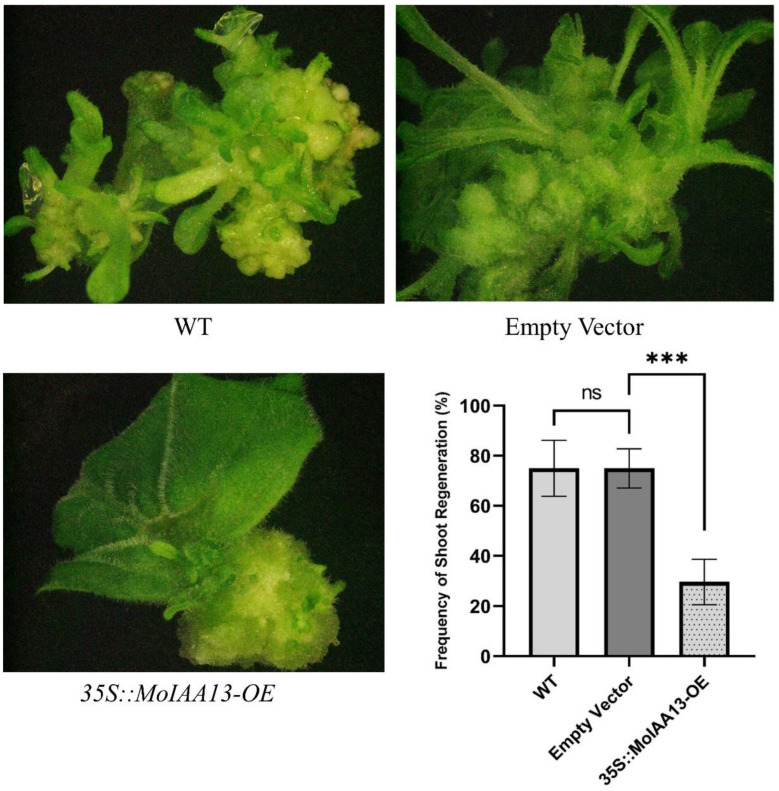
Overexpression *MoIAA13* in *N. benthamiana* compromised the shoot regeneration capacity. *** *p* < 0.001 between the *EV* (empty vector) and *35S::MoIAA13-OE* transgenic lines and *ns* represent no significance.

**Table 1 ijms-23-15729-t001:** Molecular characteristics of *MoIAA* genes in *M. oleifera*.

Name	Gene ID	Protein Length (aa)	pI	MW (kDa)
*MoIAA1*	lamu_GLEAN_10000198	135	5.83	14.69
*MoIAA2*	lamu_GLEAN_10001577	340	6.61	36.65
*MoIAA3*	lamu_GLEAN_10002613	159	9.54	17.24
*MoIAA4*	lamu_GLEAN_10004873	195	6.07	21.77
*MoIAA5*	lamu_GLEAN_10005624	357	8.53	38.75
*MoIAA6*	lamu_GLEAN_10007413	437	6.25	47.64
*MoIAA7*	lamu_GLEAN_10007551	203	7.69	22.33
*MoIAA8*	lamu_GLEAN_10007552	250	7.51	27.33
*MoIAA9*	lamu_GLEAN_10009452	198	8.72	21.81
*MoIAA10*	lamu_GLEAN_10010984	308	8.37	32.42
*MoIAA11*	lamu_GLEAN_10011061	206	6.75	23.09
*MoIAA12*	lamu_GLEAN_10011168	228	8.99	25.70
*MoIAA13*	lamu_GLEAN_10011219	247	7.61	26.30
*MoIAA14*	lamu_GLEAN_10011220	194	6.44	21.54
*MoIAA15*	lamu_GLEAN_10013390	506	7.14	54.62
*MoIAA16*	lamu_GLEAN_10013926	156	4.67	17.26
*MoIAA17*	lamu_GLEAN_10014915	175	6.83	19.28
*MoIAA18*	lamu_GLEAN_10016966	237	6.98	26.85
*MoIAA19*	lamu_GLEAN_10017136	351	8.35	38.34
*MoIAA20*	lamu_GLEAN_10018461	300	7.02	31.51
*MoIAA21*	lamu_GLEAN_10018587	253	8.15	27.45
*MoIAA22*	lamu_GLEAN_10018588	244	8.73	26.97
*MoIAA23*	lamu_GLEAN_10019362	290	9.3	30.55

**Table 2 ijms-23-15729-t002:** Analysis of Ka/Ks for the MoIAAs.

Gene Pairs	Ka	Ks	Ka/Ks
lamu_GLEAN_10018587/ lamu_GLEAN_10018588	0.380914	2.277756	0.167232
lamu_GLEAN_10011219/ lamu_GLEAN_10011220	0.341988	1.902611	0.179747
lamu_GLEAN_10009452/ lamu_GLEAN_10013926	0.276881	2.515368	0.110076
lamu_GLEAN_10018461/ lamu_GLEAN_10019362	0.318359	1.354556	0.235029
lamu_GLEAN_10011168/ lamu_GLEAN_10016966	0.363811	2.463054	0.147707
lamu_GLEAN_10005624/ lamu_GLEAN_10017136	0.417748	1.235249	0.33819
lamu_GLEAN_10001577/ lamu_GLEAN_10010984	0.21185	1.35614	0.156216
lamu_GLEAN_10007413/ lamu_GLEAN_10013390	0.244356	1.289246	0.189534
lamu_GLEAN_10007552/ lamu_GLEAN_10011219	0.187953	3.299098	0.056971
lamu_GLEAN_10007551/ lamu_GLEAN_10011220	0.22218	1.130463	0.196539
lamu_GLEAN_10002613/ lamu_GLEAN_10011061	0.38721	2.214773	0.174831

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
