# Peer review of "Genome-Wide Identification and Expression Analysis of the *Aux/IAA* Gene Family of the Drumstick Tree (*Moringa oleifera* Lam.) Reveals Regulatory Effects on Shoot Regeneration"

_ijms, 2022, doi:10.3390/ijms232415729_

Round 1

Reviewer 1 Report

In the manuscript entitled “Genome-wide identification and expression analysis of the Aux/IAA gene family in drumstick tree (Moringa oleifera Lam.) reveals regulatory effects on shoot regeneration”. The authors have identified  2MoIAA genes in the drumstick tree genome. They have functionally validated the selected genes both qRT-PCR and transgenic experiment. These  identified MoIAA genes could be important for future research study. The experiment is fine and result discussion section is well written. However, the introduction section could be improved. Some additional correction are given below.

Line no.37 vitro leaves …not clear

Line no.65 may be have been…..

Line no. 253 not clear

Author Response

1. Line no.37 vitro leaves …not clear R: We have replenished the information in our revised manuscript. 2. Line No.65 may be have been….. R: We have revised it. 3. Line No.253 not clear R: We have added the details.

Reviewer 2 Report

The purpose of this work was to get useful information to help to unravel the mechanisms of the function of Aux/IAA in shoot regeneration, and establish the foundation for improved tissue culture efficiency and molecular breeding for M. oleifera.

The research is original and relevant: An original approach to the study of auxin signaling path way, the obtained results can also be used in the research of plants of other species . There the MoIAA genes were identified in the drumstick tree genome. All MoIAA genes were located based on phylogenetic evolution analysis, the gene characteristics and promoter cis-elements were also analyzed.  Obtained results establish a foundation for further research on MoIAA gene function and  provide useful information for improved tissue culture efficiency and molecular breeding.

Topic is original research.

There is no doubt about the novelty of the research:  The study shows a high level of molecular genetic analysis. 

However, I would recommend the authors:

1. In table 1, remove column 6 "Predicated location", because this information is already in the text (line 88) and therefore it is redundant there.

The 2nd photos in Figure 8 are too dark and therefore incomprehensible, it is recommended to improve their quality and indicate what is shown in the images.

3. It would be recommended to improve the section  "Conclusion" : currently, the conclusions are very short and vague, and refer only to a part of the results.

Author Response

1. In table 1, remove column 6 "Predicated location", because this information is already in the text (line 88) and therefore it is redundant there. R: We have removed the column 6 of the Table 1. 2. The 2nd photos in Figure 8 are too dark and therefore incomprehensible, it is recommended to improve their quality and indicate what is shown in the images. R: We have improved the quality of Figure 8, including improving the brightness and readability. 3. It would be recommended to improve the section "Conclusion" : currently, the conclusions are very short and vague, and refer only to a part of the results. R: We have rewritten the conclusion part.